# Effect of Molecular Weight on the Dissolution Profiles of PEG Solid Dispersions Containing Ketoprofen

**DOI:** 10.3390/polym15071758

**Published:** 2023-03-31

**Authors:** Ha Pham Le Khanh, Ádám Haimhoffer, Dániel Nemes, Liza Józsa, Gábor Vasvári, István Budai, Attila Bényei, Zoltán Ujhelyi, Pálma Fehér, Ildikó Bácskay

**Affiliations:** 1Department of Pharmaceutical Technology, Faculty of Pharmacy, University of Debrecen, Nagyerdei körút 98, 4032 Debrecen, Hungary; 2Doctorate School of Pharmaceutical Sciences, University of Debrecen, Nagyerdei körút 98, 4032 Debrecen, Hungary; 3Institute of Healthcare Industry, University of Debrecen, Nagyerdei körút 98, 4032 Debrecen, Hungary; 4Faculty of Engineering, University of Debrecen, Ótemető utca 2-4, 4028 Debrecen, Hungary; 5Department of Physical Chemistry, Faculty of Sciences and Technology, University of Debrecen, Egyetem tér 1., 4032 Debrecen, Hungary

**Keywords:** solid dispersion, polyethylene glycol, PEG, ketoprofen, solubility, dissolution

## Abstract

Solid dispersions are typically binary systems with a hydrophilic matrix polymer and a lipophilic active substance. During formulation, the drug undergoes a crystalline to amorphous phase transition, which leads to a supersaturated solution providing enhanced bioavailability. The interaction of the active substance and the polymer is unique and influences the level of supersaturation. We aimed to investigate the relationship between low molecular weight polyethylene glycol derivates PEG 1000, 1500, and 2000 and ketoprofen regarding the effect of molecular weight. The physicochemical properties of solid dispersions prepared with hot melt homogenization and their respective physical mixtures were investigated with Fourier transform infrared spectroscopy, powder X-ray diffraction and scanning electron microscopy techniques. A phase solubility study was carried out in hydrochloric acid media which showed no difference between the three polymers, but the dissolution curves differed considerably. PEG 1000 had higher percentage of released drug than PEG 1500 and 2000, which had similar results. These results indicate that when multiple low molecular weight PEGs are suitable as matrix polymers of solid dispersions, the molecular weight has only limited impact on physicochemical characteristics and interactions and further investigation is needed to select the most applicable candidate.

## 1. Introduction

Ketoprofen, also known as 2-(3-benzoylphenyl)-propionic acid, belongs to the propionic acid group of a non-steroidal anti-inflammatory drugs (NSAID) [1]. Marketed since 1973, multiple clinical uses of ketoprofen can be listed, such as osteoarthritis, rheumatoid arthritis, gout, traumatic soft tissue injuries, low back pain, post-operative pain, headache, toothache, and fever [2,3,4]. However, the S-isomer is the more potent one compared to the R-isomer, and multiple reviews state that the racemic mixture, which is cheaper to manufacture, is still more potent than several other NSAIDs [5,6]. It is a white, crystalline powder with logP value of 0.97, melting point of 94.5 °C, and thermal stability up until 250 °C [7,8]. The Biopharmaceutics Classification System (BCS) is a system that categorizes an active pharmaceutical ingredient (API) based on its aqueous solubility and its permeability across intestinal membrane into four classes [9]. Ketoprofen belongs to BCS class II, which means it has high permeability but low water solubility, hence it has low bioavailability. Pure ketoprofen powder has around 27% gastric absorption which for example could be increased to 37% by proliposomal formulation [10]. However, the jejunum and the colon are also important sites of absorption, where the absorption is pH-dependent as pH 6.5 resulted in higher plasma levels of ketoprofen compared to 7.4 in both anatomical locations [11]. The apparent permeation coefficient is also pH dependent, having a maximum at the p*K*a value of the molecule, which is pH 4.45 [12]. As a widely used and easy to detect model API, multiple publications can be found about the enhanced oral delivery of ketoprofen, in form of liquid self-nanoemulsifying drug delivery systems, nanoscale milled powders, or salification [13,14,15].

Since their first mention in 1961, numerous studies have investigated the potential use of solid dispersions [16]. Amorphous solid dispersions (SD) are made from a BCS class II or IV API dispersed into a carrier matrix (usually a hydrophilic polymer). During the manufacturing process, the crystalline API undergoes a phase transition into an amorphous form through solvent evaporation or hot melt extrusion or spray drying to name some more common formulation methods [17,18]. The higher free energy of the amorphous drug and the nanosized particles greatly enhances the poor water solubility of the substance in gastric or intestinal fluids. The carriers can either be amorphous ones such as polyvinylpyrrolidone (PVP) or hydroxypropyl methylcellulose (HPMC), semi-crystalline ones such as polyethylene glycols (PEG), or rarely crystalline ones such as different sugars [19,20]. It was noted for the hot melt extrusion method that upon cooling, both the crystallization parameters of the API can be influenced by the polymers and the API can influence the properties of the polymer as well [21,22]. Moreover, the bioavailability can be further improved with the addition of surfactants, to the dispersion as well [23]. Even for simple systems, such as irbesartan-HPMC, the physical mixture of the two substances caused a seven-fold solubility increase, while the SD a fifteen-fold one [24]. A cyclosporine A-polyoxyethylene (40) stearate system showed ~10% dissolution for pure API, ~60% in case of the physical mixture and over 90% for the solid dispersion [25].

In conclusion it can be said that the type of polymer heavily influences the dissolution of the drug from solid dispersions. It must also be noted that the physicochemical characteristics of a polymer are directly linked to its length or molecular weight. Thus, it is logical to assume that if solid dispersions which differ only in the molecular weight of the polymer are compared to each other, the amount of released drug and the molecular weight will be proportional or inversely proportional. However, Table 1 shows that in the scientific literature, contradictory results can be found regarding the relationship of the percentage of released drug and the influence of molecular weight in case of PEGs. It can be stated that for each and every drug–PEG interaction, an individual investigation is needed to assess the exact influence of the molecular weight on the dissolution.

Numerous publications have described the use of polymers, such as PVP K30/d-mannitol or polyethylene oxide or PEG 15,000, to increase the solubility and dissolution of ketoprofen [36,37,38]. It was reported that different solid dispersions formulated with different length polymers such as Klucel™ EL or ELF and HPMC K15M and K100M had different dissolution profiles, but overall identical dissolution [39,40]. As assumed, the solid dispersion made with the higher molecular weight polymer released the ketoprofen more slowly. The aim of this study was to formulate binary ketoprofen—PEG solid dispersions with hot melt homogenization and investigate how the physicochemical characteristics and the dissolution profiles are modified due to the altering length of the matrix molecules. The primary focus was on the influence of the molecular weight and as such no other excipients were used, and the API-polymer ratio was set to 9:1 in order to further enhance the effect of the polymers on the final formulations. Physical mixtures of the API and the polymers were also tested to be compared to the dissolution profiles of the solid dispersions.

## 2. Materials and Methods

### 2.1. Materials

Three different polyethylene glycol derivatives with increasing molecular weight were purchased for the study. The nominal average molecular weight, the actual molecular weight range, the physical form of the derivates and the manufacturer are the following for the compounds: PEG 1000, 950–1050 MW, solid flaskes, (Alfa Aesar, Karlsruhe, Germany), PEG 1500, 1400–1600 MW, powder, (Molar Chemicals, Halásztelek, Hungary), PEG 2000, 1900–2200 MW, powder, (Merck Lifesciences, Budapest, Hungary). Ketoprofen (≥98%) was purchased from TCI (Zwijndrecht, Belgium).

All materials were stored under dry, cool, and dark conditions until the measurements and the sample preparation.

### 2.2. Methods

#### 2.2.1. Phase Solubility

The phase solubility test was performed by adding a fixed, excess amount of ketoprofen powder (20 mg) to 1 mL solutions of the different molecular weight PEGs at increasing concentrations (1, 5, 10, and 20 *w*/*w*%). The dissolution media was hydrochloric acid media recommended by the Ph. Eur. (425 mL of 0.2 M hydrochloric acid mixed with 250 mL 0.2 M sodium chloride and diluted to 1 L with water, pH checked and if needed adjusted to 1.2). The vials were first vortexed for 30 s to disperse the ketoprofen and then were rotated for 24 h at room temperature at 50 rpm on an Ika Loopster digital rotating shaker while being protected from light. After mixing, each vial was centrifuged at 4500 rpm for 20 min to separate the remaining solid ketoprofen particles from the saturated solutions. The samples were taken from the clear supernatant and filtered through a 0.2 µm polyethersulfone membrane filter. The ketoprofen content of the samples was analyzed by a UV spectrophotometer (Shimadzu UV-1900) at 258 nm. Previously a calibration curve was produced with serial dilution of a 40 µg/mL ketoprofen stock solution dissolved in the hydrochloric acid media. The R^2^ value of the plotted curve was 0.9913. The phase solubility profiles were plotted as the solubility of ketoprofen versus the *w*/*w*% concentration of the given PEG by GraphPad Prism (version 9; GrapPad Software, San Diego, CA, USA).

The Gibbs free energy of transfer (ΔG^0^_tr_) of ketoprofen from pure water to the aqueous solutions of the PEGs was calculated as the following:
ΔG^0^_tr_ = −2.303 × R × T × log (Ss/S_0_)
(1)

where Ss/S_0_ is the ratio of molar solubility of ketoprofen in water (41 µg/mL according to our measurements) to that in the aqueous solutions of the PEGs. The apparent stability constant (*Ka*) was determined as:
*Ka* = Slope/(Intercept × (1 − Slope))
(2)

where slope and intercept were obtained from the plotted curve.

#### 2.2.2. Preparation of Solid Dispersions and Physical Mixtures

The PEG and the ketoprofen were measured in a ratio of 9:1. We chose this ratio according to the literature results, because in this concentration, the properties of the polymer are the dominant factors of dissolution, not that of the API [20]. Solid dispersions were prepared by melting the PEGs completely at 65 °C by using the magnetic stirrer and heater Arex 6 with a thermometer probe (VELP Scientifica Usmate, Usmate Velate, Italy). Then the API was added into the PEG solution with continuous stirring. When the ketoprofen was totally dissolved in the melted PEG, thus a true solution was created, the mixture was poured into a plastic basin to cool down and solidify on room temperature. The samples were then frozen at −80 °C and were ground with a precooled mortar and pestle with continuous liquid nitrogen cooling. For the preparation of the physical mixtures, the components were measured at room temperature, and ground under the same conditions. The use of liquid nitrogen was necessary because in our preliminary studies, the PEG 1000 could form partial solid dispersions during grinding. Both samples were stored at 2–8 °C until further tests.

#### 2.2.3. Fourier-Transform Infrared Spectroscopy (FT-IR)

The infrared spectra of the pure PEGs, ketoprofen, physical mixtures, and solid dispersions were obtained by using a JASCO FT-IR 4600 type (ABL&E-JASCO, Budapest, Hungary) apparatus coupled with a Zn/Se ATR PRO ONE Single-Reflection ATR accessory. Each sample was directly placed on the cleaned crystal of the equipment; the scanning was run for 24 times in the wavelength range of 500–4000 cm^−1^ at a resolution of 1 cm^−1^ to obtain a smooth spectrum. Corrections of environmental CO_2_ and H_2_O used the built-in method of the software. Spectra were evaluated as described in our previous research [41].

#### 2.2.4. Powder X-ray Diffraction (PXRD)

The finely powdered samples were fixed onto a Mitegen MicroMeshes sample holder (MiTeGen Co., Ithaca, NY, USA) with a minimal amount of oil. Powder diffraction data of the samples with Debye Scherer geometry were collected using a Bruker-D8 Venture (Bruker AXS. GmbH, Karlsruhe, Germany) diffractometer equipped with INCOATEC IμS 3.0 dual (Cu and Mo) sealed tube micro sources (50 kV, 1.4 mA). A Photon 200 Charge-integrating Pixel Array detector and CuKα (λ = 1.54178 Å) radiation was applied. Several frames were collected with various detector-sample distances in phi rotation scanning mode. Data collection and integration were carried out using the APEX3 and DiffracEva software (Bruker AXS Inc., Madison, WI, USA, Version 4.2.2.3), respectively.

#### 2.2.5. Scanning Electron Microscopy (SEM)

Surface area exploration used a Hitachi Tabletop microscope (TM3030 Plus, Hitachi High-Technologies Corporation, Tokyo, Japan) in high-resolution mode. The samples were attached to a fixture with a double-sided adhesive tape containing graphite. Before SEM examination, gold-sputtered coating was not deposited on the surface of the samples, as the instrument is suitable for the direct investigation without any surface pre-treatments. The measurement was carried out under vacuum and low, 5 kV accelerating voltage. The magnification was 500×.

#### 2.2.6. In Vitro Dissolution Test

During the experiment, three parallel measurements were performed with size 0 capsules filled with 100 mg of the samples, meaning 90 mg of the given PEG and 10 mg of ketoprofen in form of either a solid dispersion or a physical mixture. The dissolution media was 450 mL of hydrochloric acid media (same as the media of the phase solubility study). The volume was set as the half of the maximum recommended USP volume to stay within the range of the limit of UV spectrophotometric detection of the ketoprofen. USP 2 rotating paddle method was with the rotation speed of 75 rpm and at 37 °C in an Erweka DT 128 light dissolution tester (Erweka GmbH, Langen, Germany). Samples of 1 mL were withdrawn after 5, 10, 15, 20, 30, 45, and 60 min and filtered through a 0.2 µm polyethersulfone membrane filter and measured as in case of the phase solubility study. The graphs were plotted by GraphPad Prism 9, all calculations were executed in Microsoft Excel.

In order to compare the dissolution profiles of the different formulations similarity (f1) and difference factors (f2) were calculated, as a model independent approach [42]. The exact calculations were the following:(3)$$f1=\frac{\sum_{j=1}^{n}\left|R_{j}-T_{j}\right|\;\;}{\sum_{j=1}^{n}R_{j}}\times 100$$
(4)$$f2=50\times \log\left\{\;{\left[1+\left(1/n\right)\overset{n}{\underset{j=1}{\sum }}w_{j}{\left|R_{j}-T_{j}\right|}^{2}\;\right]}^{-0.5}\times 100\;\right\}$$
where f2 is the similarity factor (the logarithmic reciprocal square root transformation of the sum of squared error is a similarity factor which provides percentage dissolution between the two curves) and *w_j_* is an optional weight factor which was 1 in our experiment.

For the fitting of dissolution profiles on different dissolution models (Table 2) the following calculations were used:

## 3. Results

### 3.1. Phase Solubility

Figure 1 demonstrates the solubilization effect of PEGs on ketoprofen. We have found that the solubility of ketoprofen in the hydrochloric acid media was 41 µg/mL. 1 *w*/*w*% PEG had essentially no effect on the solubility of the API, but for further concentrations it can be said that a linear correlation can be observed up to 10 *w*/*w*%, which drastically increased at 20 *w*/*w*% concentration. No significant differences could be found among the different PEGs in terms of their solubility-increasing effect, Table 3. The calculated apparent stability constants of ketoprofen (*Ka*) for PEG 1000 was 3.317 × 10^−5^, for PEG 1500 it was 3.346 × 10^−5^, and it was 3.569 × 10^−5^ for PEG 2000.

### 3.2. Fourier-Transform Infrared Spectroscopy (FT-IR)

Ketoprofen was formulated in two ways for further experiments. Either as a solid dispersion by hot melt homogenization (SD) or as gently mixed powders (physical mixture, PM) of ketoprofen and PEGs were tested. FT-IR spectroscopy measurements were carried out to detect any interaction between the ketoprofen and PEGs in the different formulations. In the range of 1650–1700 cm^−1^ ketoprofen had characteristic peaks which the PEGs lacked. In the case of PEG 1000 (Figure 2A), the solid dispersion and the physical mixture have nearly identical spectra. In case of PEG 1500 (Figure 2B) and 2000 (Figure 2C), the characteristic peaks of ketoprofen can be detected in the physical mixture in the same range as in the pure API. However, for the solid dispersions, the formation of a new H-bond shifted the ketoprofen bands from 1693 cm^−1^ to 1732 cm^−1^. This proves the amorphous form and the dissolution of the ketoprofen in the polymer matrix.

### 3.3. Powder X-ray Diffraction (PXRD)

The PXRD results (Figure 3) confirms that in the case of PEG 1500 and 2000, the physical mixture contains ketoprofen in a crystalline form, while in the case of solid dispersions, no suggestive signals of a crystalline structure can be found. In the solid dispersions, the active ingredient is present mostly in amorphous form which means the API was dissolved in the PEG. As compared to the FT-IR results, again, the PEG 1000 PM and the PEG 1000 SD are nearly identical, thus a partial solid dispersion is present in the physical mixture.

### 3.4. Scanning Electron Microscopy (SEM)

Figure 4 shows the SEM images of pure ketoprofen, and the three type of PEGs. In the center of Figure 4D, the typical crystalline form of ketoprofen can be seen with particle size below ~100 µm. None of the polymers show crystalline form, they are amorphous particles. The comparison of the formulated solid dispersions and physical mixtures (Figure 5) clearly indicate that the crystalline form of the active ingredient can be found between the larger particles of the PEG 2000 and 1500 for the physical mixtures, while in the case of solid dispersions we can see a molten, mostly homogeneous substance. However, for the physical mixtures of PEG 1000, the individual crystals of ketoprofen cannot be detected, and it is nearly identical to its respective solid dispersions. The absence of ketoprofen crystals confirms the formation of a solid dispersion. In the case of PEG 1000, the low melting point results in solid dispersion because ketoprofen is already in solution during gentle homogenization of the experiment.

### 3.5. In Vitro Dissolution Test

The results of the dissolution test are shown on Figure 6. The theoretical amount of dissolved ketoprofen based on the results of the phase solubility experiment (Figure 1) is presented with a dotted line. As such, during the 60 min of the experiment, supersaturated solutions of ketoprofen could be detected in all cases. The highest amount of dissolved drug could be detected for PEG 1000, while the formulations with the other two PEGs had mostly similar amount of dissolved ketoprofen. Difference and similarity factors of the dissolution profiles were also calculated in Table 4. If f1 values are between 0 and 15 and f2 values are above 50, then the two compared curves are statistically similar. It can be seen on the graph that PEG 1500 SD and 2000 SD are similar to each other.

Further analysis of the dissolution curves can be seen in Table 5. Most samples had the best fitting with the Korsmeyer–Peppas model, which is used to describe the release from polymeric systems [43]. However, in certain cases, zero-order model was the most fitting, which is also applicable for matrix tablets with low solubility API. Overall, all formulations showed a delayed, prolonged release compared to the first-order kinetic.

## 4. Discussion

Ketoprofen is a widely used NSAID with multiple therapeutical indications, yet with an extremely low water solubility. In order to increase its bioavailability, this low solubility must be improved which can be achieved through various methods. The application of solid dispersion is an increasingly popular way to formulate poor solubility drugs because of the advantageous properties of the final product [45]. In our study, we aimed to investigate binary dispersions of ketoprofen with three different molecular weight polymers, PEG 1000, 1500, and 2000 to highlight the effect of molecular weight on the manufactured solid dispersions and their dissolution characteristics.

Our phase solubility study (Figure 1) has revealed a moderate increase in PEG solubility. We have found that the solubility of ketoprofen was 41 µg/mL in hydrochloric acid media, which was increased to 300 µg/mL in the presence of 20 *w*/*w*% PEG. There was no statistical difference between the three polymers. This is similar to the results of Khattab et al. who found only limited difference between the solubility increasing capabilities of PEG 4000, 10,000 and 20,000 for gliclazide [34]. Our results in the 1–10 *w*/*w*% range correlate well with the previous findings of Mura et al. who reported that in this range, the PEGs increased the solubility of ketoprofen proportionally [38]. Table 3 shows negative free energy transfer for all concentrations and polymers, demonstrating spontaneity of the drug solubilization process. The decreasing free energy for the higher concentration solutions indicate that this process is even more favorable at higher concentrations of the polymers. According to the *Ka* values, the binding affinity between the ketoprofen and the PEGs increase with the molecular weight.

When comparing the six samples, both the FT-IR (Figure 2) and the PXRD (Figure 3) results prove that even on room temperature, a solid dispersion is formed in case of the physical mixture of PEG 1000 and ketoprofen. This low melting range is the key reason why PEG 1000 is rarely used for the preparation of hot melt solid dispersions. The observed shifts of the characteristic IR-bands are caused by the formation of H-bonds between the ketoprofen and the polymer. This phenomenon and the peak intensity changes in case of the PXRD were reported previously for different ketoprofen solid dispersions [19,46,47]. In general, such changes of the respective spectra are the clear signs of the formation of a solid dispersion from a simple physical mixture [48,49]. The homogenous melted structure of the solid dispersion is further confirmed by the scanning electron microscopy images (Figure 5B,D,F). The PEG 2000 and PEG 1500 physical mixtures showed the individual ketoprofen crystals, while the PEG 1000 physical mixture was identical to its solid dispersion counterpart (Figure 5A,C,E). Dabbagh et al. investigated the phase transition of the three polymers through differential scanning calorimetry [50]. They found that the melting starts from 25.3 ± 2.5 °C in case of PEG 1000, 40.4 ± 1.8 °C for PEG 1500, and 45.4 ± 1.5 °C for PEG 2000. The peak phase transition temperature for PEG 1000 was 37.1 ± 2.3 °C, and for the two other polymers it was 48.3 ± 1.7 °C and 50.0 ± 1.3 °C respectively. This low melting point of the PEG 1000 explains the formation of solid dispersions even at room temperature and the lack of such phenomenon for the other two polymers.

The main advantage of solid dis”Irs’on is their ability to form supersaturated solutions of the given API. In an ideal system, the 100% of the API is in amorphous state, thus, the dissolution of the matrix polymer will result in a supersaturated colloidal solution, in which the kinetic solubility is multiple times higher than the thermodynamic solubility which depends on the physicochemical properties of the crystalline form [51]. The difference between the two solubility values can be observed when comparing Figure 1, Figure 2, Figure 3, Figure 4, Figure 5 and Figure 6. During the 24 h of the phase solubility study, an equilibrium was reached between the crystalline form of the ketoprofen and the different concentrations of PEGs in the hydrochloric acid media. The undissolved ketoprofen crystals remained in the solutions as solid particles [52]. Meanwhile during the dissolution experiment, the time window was enough short for the detection of the supersaturated solution be detected but too short to reach a new equilibrium. On Figure 6, the dotted line represents the theoretical maximum of how high the solubility of ketoprofen should have been in the presence of 90 mg PEG in 450 mL of hydrochloric acid media. This supersaturation is explained by temporary prevention of crystallization by the polymer matrix [53,54,55]. Thus, the possibly enhanced bioavailability can be assumed [24,56,57].

Discussing the contradictory results of Table 1, a possible explanation is given by Duong and Van den Mooter [20]. In their review they claim that because fenofibrate does not interact with PEG through the formation of hydrogen bonds, while flurbiprofen has such secondary bonding, the dissolution of the solid dispersion is not affected by molecular weight for fenofibrate [58,59]. However, it was found that for the possible hydrogen bond-forming molecule naproxen the molecular weight had no impact on the dissolution profiles of the solid dispersions. We also found that in the relationship of ketoprofen and PEG 1000, 1500, and 2000, despite the proven formation of hydrogen bonds, there was no direct relationship between the molecular weight and the dissolution properties.

The analysis of the dissolution curves shows interesting phenomena. On the one hand, both the physical mixtures and the solid dispersions of PEG 1500 and 2000 had significantly similar results (Figure 6, Table 4 and Table 5). A possible explanation can be that the dissolution speed of the polymer can be decreased if it is a part of a solid dispersion and not freely present as in a physical mixture [60]. Moreover, the binding affinity of ketoprofen toward the two polymers are higher than in case of the PEG 1000, which means that the release can be slower, despite the amorphous state of the API.

On the other hand, it can be observed that for the PEG 1000, the solid dispersion dissolved significantly higher amount of ketoprofen than its respective physical mixture. Based on the literature results, it can be said for sure that during the experiment, PEG 1000 was melting in each capsule at a temperature of 37 °C. However, in the solid dispersion, ketoprofen was already present in an amorphous state, while in the physical mixture, the phase transition was just beginning, thus an in situ solid dispersion was created. The results can only be explained by the contrary effects of the solubilization of crystalline ketoprofen by PEGs, the melting of PEGs inside the capsules, and the dissolution speed of PEGs in solid dispersions and in physical mixtures.

Our finding of the kinetics profile (Table 4) for the samples are also confirmed by the literature, as solid dispersions tend to fit well for both the Korsmeyer–Peppas and the zero-order model as well [39,61,62].

Overall, it was proved that in case of low molecular weight PEGs, the length of the polymer has no direct influence on the dissolution profile of ketoprofen solid dispersions. Unfortunately, our results can only be compared to other research articles moderately. The investigation of binary solid dispersions are out of scope nowadays for researchers, and most cited publications did not focus on the impact of molecular weight but other issues of solid dispersion formulation. Further research is needed to study the dissolution speeds of the different formulations in the target media to explain the interesting results of the dissolution study. Only the PEG 1000-ketoprofen solid dispersion was superior to its physical mixture counterpart, while PEG 1500 and 2000 were very similar to each other. Moreover, the physical mixture of PEG 1000 had relatively high dissolved amount of API. These results indicate that in case of low molecular weight PEGs, the preparation of a solid dispersion might not be needed to achieve a supersaturated solution in the stomach and as such, drastically increase the bioavailability of the given API. Even the simple physical mixture can improve drug delivery due to the combination of the melting point and dissolution speed of the PEGs and the unique interaction between the ketoprofen and the polymers. Further research is needed to investigate the possible use of this feature of low molecular weight polyethylene glycol derivatives.

## 5. Conclusions

We report the preparation of ketoprofen–PEG physical mixtures and solid dispersions in 1:9 ratio. We found that at room temperature, PEG 1000 physical mixture started melting, thus creating a partial solid dispersion, while in case of PEG 1500 and 2000, the two formulations had different PXRD and FT-IR spectra and their physical appearance differed according to the scanning electron microscopic images. The dissolution study carried out in hydrochloric acid media showed no difference between the PEG 1500 and 2000 formulations and revealed significantly higher amount of dissolved API in case of PEG 1000 solid dispersion. However, in all cases a supersaturated solution was detected as the amount of dissolved ketoprofen exceeded the theoretical maximum, indicated by the phase solubility study.

## Figures and Tables

**Figure 1 polymers-15-01758-f001:**
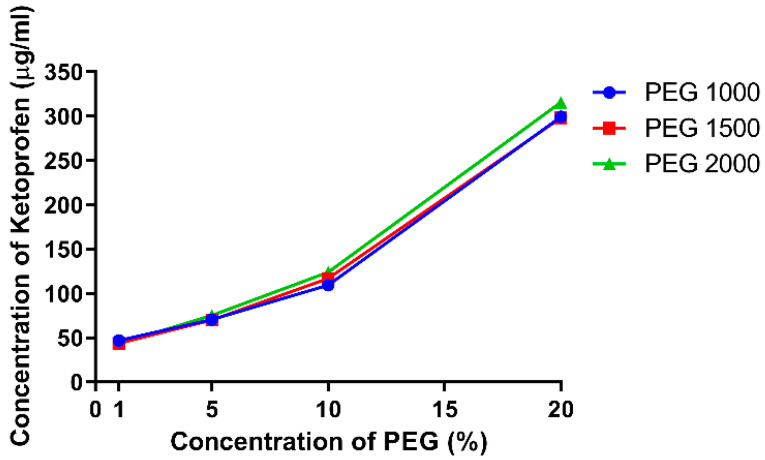
Phase solubility diagram of ketoprofen in hydrochloric acid media at 37 °C in the presence of different PEGs. Data expressed as mean ± SD, *n* = 3.

**Figure 2 polymers-15-01758-f002:**
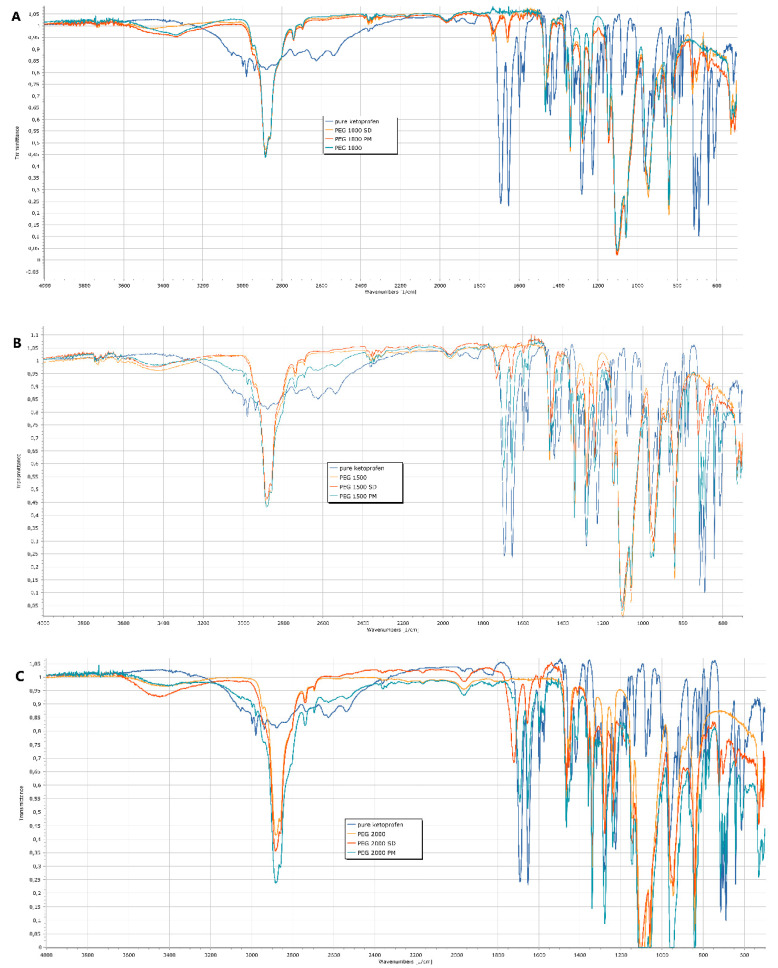
FT-IR spectra of ketoprofen, pure PEGs, physical mixtures (PM), solid dispersions (SD) for PEG 1000 (**A**), PEG 1500 (**B**), and PEG 2000 (**C**). The intensity peaks were normalized for the highest intensity peak.

**Figure 3 polymers-15-01758-f003:**
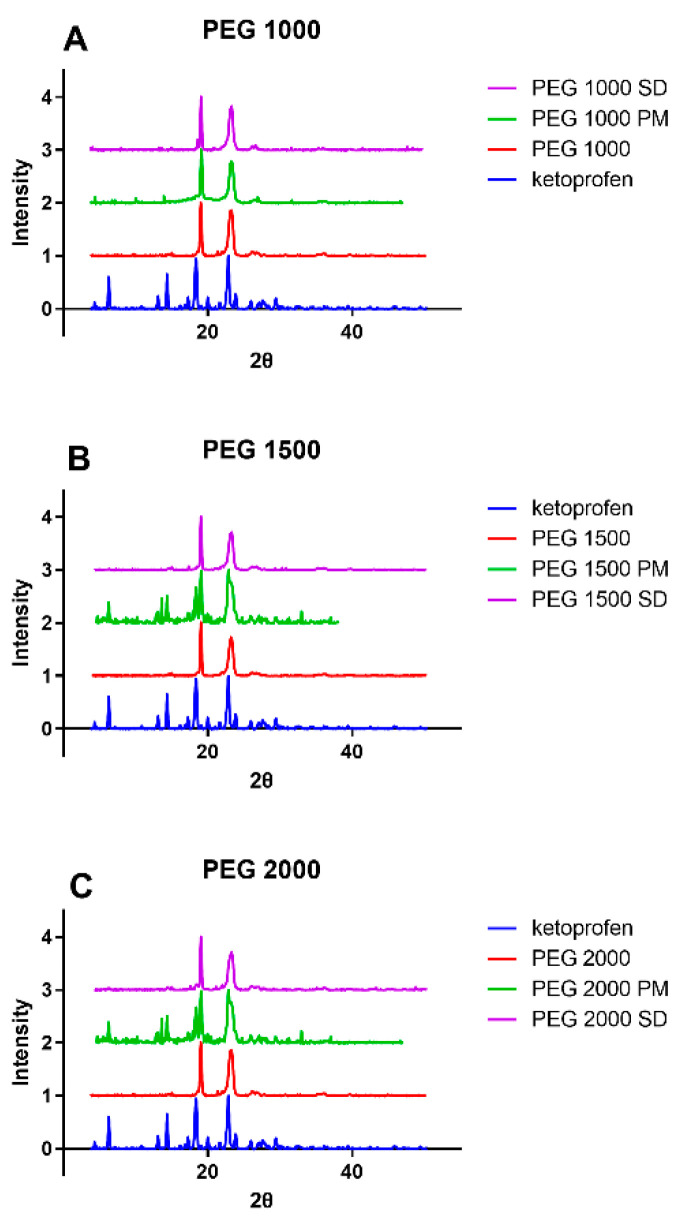
PXRD spectra of ketoprofen, pure PEGs, physical mixtures (PM), solid dispersions (SD) for PEG 1000 (**A**), PEG 1500 (**B**), and PEG 2000 (**C**).

**Figure 4 polymers-15-01758-f004:**
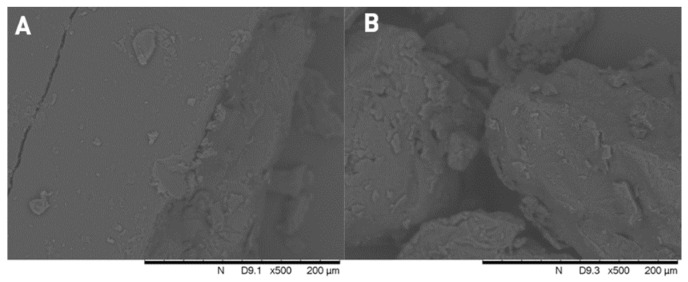
Scanning electron microscope images of PEG 2000 (**A**), PEG 1500 (**B**), PEG 1000 (**C**), and crystalline ketoprofen (**D**).

**Figure 5 polymers-15-01758-f005:**
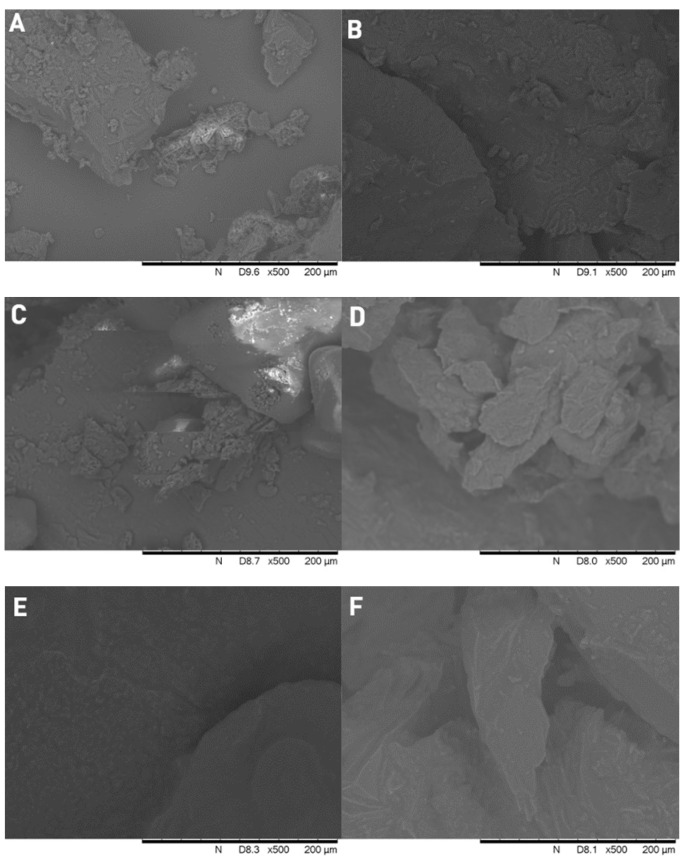
Scanning electron microscope images of PEG 2000 physical mixture (**A**) and solid dispersion (**B**); PEG 1500 physical mixture (**C**) and solid dispersion (**D**); PEG 1000 physical mixture (**E**) and solid dispersion (**F**).

**Figure 6 polymers-15-01758-f006:**
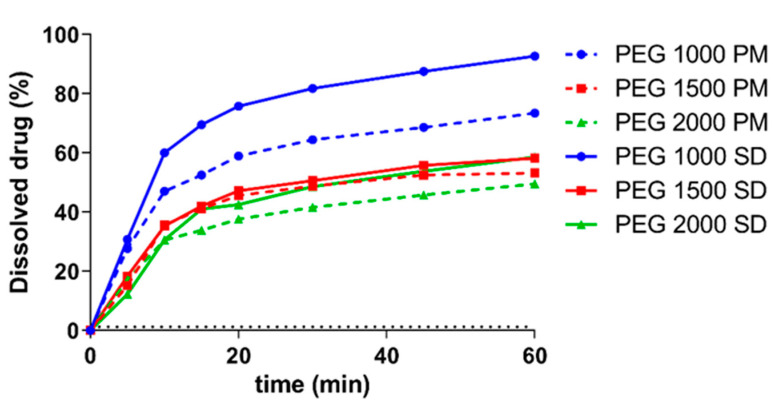
Dissolution profiles of solid dispersions and physical mixtures in hydrochloric acid media. Data are represented as the average (±SD) detected amount of ketoprofen compared to the average amount of ketoprofen loaded into the capsules, *n* = 3. The dotted line represents the theoretical maximum solubility of ketoprofen in the presence of the PEGs based on the phase solubility study (90 mg of PEG in 450 mL of hydrochloric acid media, average is calculated and plotted from the result of the three polymers).

**Table 1 polymers-15-01758-t001:** Literature review of dissolution studies which involved solid dispersions, prepared with multiple PEGs.

Publication	API	PEG	Relationship of Molecular Weight and Percentage of Released Drug
Ford et al. [26]	indomethacin, phenylbutazone	1500, 4000, 6000, 20,000	Inverse proportionality
Miralles et al. [27]	tolbutamide	4000, 6000	Inverse proportionality
Serajuddin et al. [28]	α-Pentyl-3-(2-quinolinylmethoxy)benzene methanol	1000, 1450, 8000	Inverse proportionality
Chiou and Riegelman [29]	griseofulvin	4000, 20,000	Proportional
Betageri and Makarla [30]	glyburide	4000, 6000	Proportional
Anguiano-Igea et al. [31]	clofibrate	10,000, 20,000, 35,000	Proportional
Dordunoo and Rubinstein [32]	triamterene, temazepam	1500, 2000, 4000, 6000	Proportional
Mura et al. [33]	naproxen	4000, 6000, 20,000	None
Khattab et al. [34]	gliclazide	4000, 10,000, 20,000	None
Trapani et al. [35]	zolpidem	4000, 6000	None

**Table 2 polymers-15-01758-t002:** Mathematical calculations of drug release profiles.

Model Name	Equations [43,44]	GRAPHIC
Zero-order	$Q_{t}=Q_{0}+k_{0}t$	The graphic of the drug-dissolved fraction versus time is linear.
First-order	$Q_{t}=Q_{0}\times e^{-k_{1}t}$	The graphic of the decimal logarithm of the released amount of drug versus time is linear.
Korsmeyer–Peppas model	$\frac{Q_{t}}{Q_{\infty }}=k_{kp}t^{n}\left(\mathrm{up}\;\mathrm{to}\;\frac{Q_{t}}{Q_{\infty }}\geq 0.6\right)$	The graphic of the released drug versus the square root of time should form a straight line.

where *Q* is the amount of drug release at time *t, Q_0_* is the initial amount of drug, *Qt* is the amount of drug remaining at time *t*, and where *Q_t_/Q_∞_* is fraction of drug released at time *t*. *k*_0_, *k*_1_, and *k_kp_* are the kinetic constants for zero-order, first-order, and Korsmeyer–Peppas models, respectively and *n* is the release exponent, indicative of the drug release mechanism.

**Table 3 polymers-15-01758-t003:** Gibbs free energy values of ketoprofen and the tested PEGs.

Concentration (*w*/*w*%)	ΔG^0^_tr_ (J/mol)		
	PEG 1000	PEG 1500	PEG 2000
1	−3316	−1445	−1166
5	−1342	−1316	−1491
10	−2420	−2585	−2731
20	−4911	−4901	−5042

**Table 4 polymers-15-01758-t004:** Difference and similarity factors of the dissolution profiles of the samples.

Comparison of Compositions	f1	f2
PEG 1000 PM vs. SD	29.80	39.84
PEG 1500 PM vs. SD	**5.00**	**76.78**
PEG 2000 PM vs. SD	**10.84**	**59.19**
PEG 1000 SD vs. PEG 1500 SD	38.27	25.74
PEG 1000 SD vs. PEG 2000 SD	42.30	27.59
PEG 1500 SD vs. PEG 2000 SD	**6.54**	**71.39**

**Table 5 polymers-15-01758-t005:** Coefficient determination (R^2^) for different models of the dissolution profiles of the samples.

Type of Kinetics Profile		Zero-Order Model	First-Order Model	Korsmeyer–Peppas Model
Physical mixture	PEG 1000	0.8440	0.8854	**0.9321**
	PEG 1500	**0.8371**	0.7104	0.7859
	PEG 2000	0.8588	0.8715	**0.9109**
Solid dispersions	PEG 1000	0.8416	0.9469	**0.9592**
	PEG 1500	0.8547	0.8242	**0.8710**
	PEG 2000	**0.8823**	0.8543	0.8444

## Data Availability

The data presented in this study are available on request from the corresponding author.

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
