# Peer review of "Effect of Molecular Weight on the Dissolution Profiles of PEG Solid Dispersions Containing Ketoprofen"

_polymers, 2023, doi:10.3390/polym15071758_

Round 1
Reviewer 1 Report
Dear Sir,
The work titled "Effect of molecular weight on the dissolution profiles of PEG solid dispersions containing ketoprofen". The researchers investigated the effect of molecular weight of PEG 1000, 1500 and 2000 on the dissolution curves of ketoprofen solid dispersions.
The comments are as follows:
1. How this research work is different from available literature on PEG-based solid dispersions? What is the novelty here?
2. Why is PEG 4000 or PEG 15000 (as mentioned in line 71) not tried for comparison purposes?
3. Abstract lines 26-29: These results highlight, that individual consideration is needed in the selection of polymer for solid dispersions as relatively similar polymers can drastically modify the pharmacokinetics of the final product.
Is this talking about PEG?
4. Line: 34 to 37. Marketed since 1973, multiple clinical uses can be listed, such as osteoarthritis, rheumatoid arthritis, gout, traumatic soft tissue injuries, low back pain, post-operative pain, headache, toothache, and fever.
Is ketoprofen missing from this sentence?
5. Line: 74-75: On the contrary, HPMC K15M and K100M had no such difference.
Is this polymer used for Solubility improvement or retarding it? Please revisit the sentence as, with this polymer as you increase MW, it will slow down the release.
6. Why fix ratio used, a ratio of 9:1? It can be 1:1, 5:1, or 10:1 also.
7. Line 117: Why was frozen with liquid nitrogen used? Normal freeing is not enough.
8. Line 130: Remove hyperlink only cite it, Germany, http://spectroscopy.ninja (accessed on 15 January 2021))
9. Saturated solubility study should be performed in target media and compared.
10. Figure 4, needs to be more clear.
11. Table 4, must have the findings of this research, for better comparison. This table should not be discussed; it should be part of the introduction.
12. Discussion needs to improve, with the addition of results finding and citing the recent work done.
Author Response
24/03/2023
Dear Reviewer 1,
I hereby submit our modified manuscript “Effect of molecular weight on the dissolution profiles of PEG solid dispersions containing ketoprofen”, by Pham Le Khanh Ha et al. Thank you for your constructive comments, the changes in the text are highlighted with cyanide color. Our answers to your comments are the following:
- How this research work is different from available literature on PEG-based solid dispersions? What is the novelty here?
The same type of polymers with different molecular weight are rarely tested under the same conditions because it is presumed, that certain properties are linked to the length of the polymer. In our previously published paper, the relationship between molecular weight and cellular effects of PEGs was investigated and we have found no correlation. (Comparative Investigation of Cellular Effects of Polyethylene Glycol (PEG) Derivatives DOI: 10.3390/polym14020279).
We planned to continue this investigation in the field of pharmaceutical formulations and then we have found lot of contradictory literature results (Table 4. in the original manuscript) about solid dispersions. As a conclusion we came to say, that no general rule can be applied, but each and every API-polymer interaction must be studied individually. Thus, the submitted manuscript is the first study, describing solid dispersions with three different PEGs as matrix polymers, prepared under the same conditions, containing ketoprofen and investigated with SEM, PXRD and FT-IR and additional solubility studies. These results can also be interesting, because most other studies involving PEGs utilize PEGs with higher molecular weight, and our study reveals that the physical mixtures of low molecular weight PEGs can provide the same amount of dissolved API as solid dispersions, creating supersaturated solutions.
- Why is PEG 4000 or PEG 15000 (as mentioned in line 71) not tried for comparison purposes?
We aimed to focus our study purely on the effect of molecular weight to find any possible correlation between the physical and dissolution properties of the prepared solid dispersions. The regular steps (~ 500 MW) between the molecular weight investigated polymers would have made it easier to interpret the results, if we had found any correlation. We tried, but could not purchase PEG 2500, to further increase the spectrum of the study.
Also, as mentioned above, most studies utilize PEGs with higher molecular weight, thus we wanted to highlight the PEGs with low molecular weight. However, for further research purposes, we aim to compare an even higher number of PEG derivatives to investigate the dissolution profiles of their physical mixtures.
- Abstract lines 26-29: These results highlight, that individual consideration is needed in the selection of polymer for solid dispersions as relatively similar polymers can drastically modify the pharmacokinetics of the final product.
Is this talking about PEG?
Yes, the sentence is modified to clarify the meaning.
- Line: 34 to 37. Marketed since 1973, multiple clinical uses can be listed, such as osteoarthritis, rheumatoid arthritis, gout, traumatic soft tissue injuries, low back pain, post-operative pain, headache, toothache, and fever.
Is ketoprofen missing from this sentence?
It is inserted into the sentence.
- Line: 74-75: On the contrary, HPMC K15M and K100M had no such difference.
Is this polymer used for Solubility improvement or retarding it? Please revisit the sentence as, with this polymer as you increase MW, it will slow down the release.
The whole paragraph is reformulated to clarify the aims and novelty of the study.
- Why fix ratio used, a ratio of 9:1? It can be 1:1, 5:1, or 10:1 also.
During literature review, we have found that 9:1 ratio is the best, if the effect of the polymer on the dissolution is the question. If the ratio of API is increased, its physicochemical characteristics will have higher impact on the dissolution profile of the solid dispersion. However we accept your suggestion and in further research involving multiple PEG derivatives, we will compare solid dispersions with different polymer:API ratios.
- Line 117: Why was frozen with liquid nitrogen used? Normal freeing is not enough.
While carrying out the experiment, we found that even if the materials were cooled to -20 or -80 °C, during grinding, the applied force melts them and a partial solid dispersion is formed in case of PEG 1000. Thus, we used continuous liquid nitrogen during the grinding to prevent any premature formation of solid dispersions in case of the physical mixtures. The explanation is added to the methods part.
- Line 130: Remove hyperlink only cite it, Germany, http://spectroscopy.ninja (accessed on 15 January 2021))
It is modified, with reference to our previous work. (Haimhoffer et al. Investigation of the Drug Carrier Properties of Insoluble Cyclodextrin Polymer Microspheres. DOI: 10.3390/biom12070931.)
- Saturated solubility study should be performed in target media and compared.
The phase solubility was performed in the same media, as the dissolution study. It is clarified in the methods and the text again.
- Figure 4, needs to be more clear.
The experiment was carried out again, all derivatives are included now in the manuscript.
- Table 4, must have the findings of this research, for better comparison. This table should not be discussed; it should be part of the introduction.
The table is moved to the introduction. The comparison of the literature results and our findings are included in the discussion.
- Discussion needs to improve, with the addition of results finding and citing the recent work done.
The discussion is rephrased again. We aimed to find all available publications on solid dispersion studies involving polymers with different molecular weight. These articles were cited and compared to our results in table 1 or in the discussion. Unfortunately, this topic is out of scope of current researchers as most of the found articles were published before 2010.
We hope that our improved manuscript is worth publication!
Waiting for your kind response.
Sincerely,
Ildikó Bácskay
corresponding author

Reviewer 2 Report
The current research manuscript investigating the effect of different grades of PEG on the release profiles of binary amorphous solid dispersions of ketoprofen is not suitable for the journal standard and lacks novelty. The manuscript lacks a proper flow of content and is written very generally. Various non-scientific terms were used. The entire manuscript needs to be updated for methodology and results. The manuscript also requires English Editing. A few of the comments are listed below:
1. It is suggested to elaborate on the abbreviations when used for the first time within the text.
2. Introduction needs to be improved. Please consider including information for solubility, melting point, degradation point, and Log P for the drug. Also, the scope of the work needs to be discussed in a more detailed manner.
3. Please mention what was the form of different PEG grades (liquid/flakes / powder).
4. What was the PSD of API?
5. Methodologies need to be described in detail. For example, the phase solubility studies need to contain detailed concentrations rather than mentioning the range, approximate amount of drug, temperature if maintained, what instrument was used, calibration curve, the filter used for collecting the samples, and correlation coefficient. All other methodologies also need to be updated.
6. Section 2.2.6: what was the dose? Why 450 mL of media was used? What size of the filter was used for collecting the samples?
7. Results need to be discussed in detail. The authors haven’t provided any information for calculating similarity factors and for different drug release models in the methodology section.
8. Line 192: Please correct the typo “Amorp”
9. The conclusion needs to be improved.
Author Response
24/03/2023
Dear Reviewer 2,
I hereby submit our modified manuscript “Effect of molecular weight on the dissolution profiles of PEG solid dispersions containing ketoprofen”, by Pham Le Khanh Ha et al. Thank you for your constructive comments, the changes in the text are highlighted with yellow color. Our answers to your comments are the following:
- It is suggested to elaborate on the abbreviations when used for the first time within the text.
All abbreviations are checked carefully in the text.
- Introduction needs to be improved. Please consider including information for solubility, melting point, degradation point, and Log P for the drug. Also, the scope of the work needs to be discussed in a more detailed manner.
The required data are now part of the introduction and the whole introduction is rephrased.
- Please mention what was the form of different PEG grades (liquid/flakes / powder).
It is now included in the methods.
- What was the PSD of API?
No particle size analysis was carried out on the API as during the preparation of the solid dispersions a true solution is created and on the SEM images, it can be seen, that the crystalline form of ketoprofen cannot be detected in the final solid dispersions. The SEM images were redone for the revised manuscript and given the scale of the images, all particles of ketoprofen were below 100 µm.
- Methodologies need to be described in detail. For example, the phase solubility studies need to contain detailed concentrations rather than mentioning the range, approximate amount of drug, temperature if maintained, what instrument was used, calibration curve, the filter used for collecting the samples, and correlation coefficient. All other methodologies also need to be updated.
All sections of Methods and materials are checked again, and additional details are given.
- Section 2.2.6: what was the dose? Why 450 mL of media was used? What size of the filter was used for collecting the samples?
Each sample was 100 mg of a 9:1 PEG:ketoprofen mixture or solid dispersion. It is clarified in the text. An explanation of the volume was added to the text also.
- Results need to be discussed in detail. The authors haven’t provided any information for calculating similarity factors and for different drug release models in the methodology section.
All calculations and the relevant publications are included in the modified section of the methods. We aimed to find all available publications on solid dispersion studies involving polymers with different molecular weight. These articles were cited and compared to our results in table 1 or in the discussion. Unfortunately, this topic is out of scope of current researchers as most of the found articles were published before 2010.
- Line 192: Please correct the typo “Amorp”
The typo is corrected. The manuscript was checked carefully for any additional typos and grammatical mistakes.
- The conclusion needs to be improved.
The conclusion and the discussion are rephrased in the resubmitted manuscript.
We hope that our improved manuscript is worth publication!
Waiting for your kind response.
Sincerely,
Ildikó Bácskay
corresponding author

Round 2
Reviewer 1 Report
Accepted in present form.
Reviewer 2 Report
All the comments are well addressed with proper justification and supporting literature. The revised version of the manuscript can be accepted for publication.